# Direct-Acting Antiviral Therapy for Hepatitis C Virus in Patients with BCLC Stage B/C Hepatocellular Carcinoma

**DOI:** 10.3390/v14112316

**Published:** 2022-10-22

**Authors:** Shou-Wu Lee, Li-Shu Chen, Sheng-Shun Yang, Yi-Hsiang Huang, Teng-Yu Lee

**Affiliations:** 1Division of Gastroenterology and Hepatology, Department of Internal Medicine, Taichung Veterans General Hospital, Taichung 40705, Taiwan; 2School of Medicine, Chung Shan Medical University, Taichung 40201, Taiwan; 3School of Medicine, National Yang Ming Chiao Tung University, Taipei 11221, Taiwan; 4Department of Post-Baccalaureate Medicine, College of Medicine, Chung Hsing University, Taichung 40227, Taiwan; 5Ph.D. Program in Translational Medicine, Chung Hsing University, Taichung 40227, Taiwan; 6Institute of Biomedical Sciences, Chung Hsing University, Taichung 40227, Taiwan; 7Division of Gastroenterology and Hepatology, Department of Medicine, Taipei Veterans General Hospital, Taipei 11217, Taiwan; 8Institute of Clinical Medicine, National Yang Ming Chiao Tung University, Taipei 11221, Taiwan

**Keywords:** cirrhosis, direct acting antiviral agents, hepatocellular carcinoma

## Abstract

Background: The benefits of hepatitis C virus (HCV)eradication for hepatocellular carcinoma (HCC) patients in Barcelona Clinic Liver Cancer (BCLC) stage B/C remain uncertain. Methods: In this hospital-based cohort study, all HCV-infected patients with BCLC stage B/C HCC during the period January 2017 to March 2021 were retrospectively screened, with 97 patients who had completed direct-acting antiviral (DAA) therapy being enrolled for final analysis. Results: In total, the sustained virological response (SVR) rate was 90.7%. In logistic regression analysis, progressive disease (PD) to prior tumor treatments was significantly associated with SVR failure (odds ratio 5.59, 95% CI 1.30–24.06, *p* = 0.021). Furthermore, the overall survival (OS) rate was significantly higher in the SVR group than that in the non-SVR group (1-year OS: 87.5% vs. 57.1%, *p* = 0.001). SVR was found to be an independent factor related to OS (hazard ratio 8.42, 95% CI 2.93–24.19, *p* = 0.001). However, even upon achieving SVR, the OS rates in BCLC stage C or Child–Pugh stage B patients remained poor. Conclusions: In BCLC stage B/C HCC, DAA could achieve a high SVR rate except in those patients with PD to prior HCC treatments. SVR was related to improvements in OS; therefore, DAA therapy should be encouraged for patients diagnosed without a short life expectancy.

## 1. Introduction

Hepatocellular carcinoma (HCC) is one of the most common causes of cancer-related death worldwide, with the causes of HCC including viral hepatitis, such as hepatitis C virus (HCV). In the era of interferon therapy for HCV treatment, the effects of interferon have been confirmed to reduce HCC recurrence rates while also prolonging survival time [1]. However, due to the prolonged antiviral therapy duration, low viral eradication rate, and severe side effects, patients with cancer and cirrhosis were often excluded from undergoing therapy during the interferon era [2].

Direct-acting antiviral agents (DAAs) are known to improve HCV infection outcomes, even in patients experiencing advanced liver disease, while also having a good safety profile as well as a sustained virological response (SVR) rate exceeding 95% in clinical practice [3]. Several studies have reported that DAA therapy reduces incidents of HCC and mortality among HCV-infected individuals [4,5,6]. However, most enrolled patients have had no history of HCC or those who were at the early stages of HCC were successfully treated, and these groups had thus been prioritized over those diagnosed with intermediate or advanced HCC, i.e., Barcelona Clinic Liver Cancer (BCLC) stage B or C. Additionally, SVR rates were lower in the patients with HCC than those without [7].

BCLC stage B/C HCC, i.e., multiple large tumors, tumors with vascular invasion, and/or extrahepatic metastasis, is basically incurable, and its patient-survival time is much shorter than that of BCLC stage A (early HCC) [8]. According to the current data, there is no clear evidence which proves that DAAs have a beneficial impact on outcomes in patients with BCLC stage B/C HCC. Nevertheless, the benefit of DAA therapy in BCLC stage B/C patients is that the natural history of these cases does not worsen, which allows for the possibility of administering anti-cancer treatment [9]. However, considering the higher mortality rate and lower SVR rate due to advanced stage liver cirrhosis and viable active tumors, the benefits of DAA therapy in this group of patients need to be clearly identified.

This study aimed to evaluate the SVR rate and its survival benefits in patients with BCLC stage B/C HCC as well as determine the factors affecting poor survival outcomes in patients with SVR following DAA therapy.

## 2. Materials and Methods

### 2.1. Patients Enrollment and Data Organization

Data on patients with HCC BCLC stage B or C at Taichung Veterans General Hospital during the period from January 2017 to March 2021 were screened retrospectively. HCC was diagnosed in accordance with the AASLD guidelines [10]. For avoiding an immortal time bias in outcome analysis, only patients who completed DAA therapy with available SVR data were included. SVR was defined as undetectable HCV RNA for at least 12 weeks (SVR12) after completion of DAA therapy.

The general data of the enrolled subjects, including age, gender, Child–Pugh stage, Albumin-Bilirubin (ALBI) grade, Fibrosis-4 (FIB-4) scores, and HCC characteristics as well as their laboratory data including serum level of bilirubin, alanine aminotransferase (ALT), alpha-fetoprotein (AFP), HCV genotype, and values of HCV viral load were all collected from the patients prior to initiating DAA therapy. Child–Pugh stage (A/B/C) was evaluated by the composite parameters (albumin, bilirubin, prothrombin time, ascites, and hepatoencephalopathy) to assess the severity of chronic liver disease [11]. ALBI grade (1/2/3) was calculated by albumin and bilirubin, which was another useful parameter to access liver dysfunction [12]. FIB-4 score was calculated by ALT/ aspartate aminotransferase/ platelet/ age, which was a non-invasive parameter to estimate the degree of liver fibrosis [13].

The HCC characteristics were reviewed by experienced radiologists according to each individual’s latest dynamic images before initiating DAA therapy. The largest tumor size, ratio of within up-to-7 criteria, and tumor response to prior HCC treatment, including complete response (CR), partial response (PR), stable disease (SD) and progressive disease (PD), based on the modified RECIST (mRECIST) criteria [14], were all recorded. The up-to-7 criteria means HCC with seven as the sum of the diameter of the largest tumor (in cm) and the number of tumors, and HCC beyond up-to-7 criteria indicates a large tumor burden [15]. The treatments implemented for HCC before and after initiating DAA therapy, including local-regional therapy (LRT), tyrosin kinase inhibitor (TKI), and immune checkpoint inhibitor (ICI), were also registered. The exclusion criteria included cases diagnosed with HCC BCLC stage A or D (i.e., early or terminal HCC) and subjects with an undetectable HCV viral load, a lack of compliance with drugs, or loss to follow-up within the following day.

The DAA regimens in our study included 24-week daclatasvir plus asunaprevir (DCV/ASV) for genotype 1b, 12-week ombitasvir and paritaprevir with ritonavir plus dasabuvir (PrOD) for genotype 1, 12-week elbasvir plus grazoprevir (EBR/GZR) for genotype 1b, 12-week sofosbuvir plus ledipasvir (SOF/LDV) for genotype 1 and 6, 12-week SOF plus RBV for genotype 2, 12-week SOF plus daclatasvir (SOF/DCV) for genotype 1 or 2, 8-week glecaprevir-pibrentasvir (G/P) for all genotypes, and 12-week sofosbuvir plus velpatasvir (SOF/VEL) for all genotypes. SOF/DCV, SOF/VEL, and G/P were defined as pangenotypic DAAs.

The subjects with SVR to DAAs were classified as the SVR group, while those without SVR to DAAs were classified as the non-SVR group. The characteristics between these two groups were compared. Overall survival (OS) was defined as the time from the start of DAA therapy until death.

### 2.2. Statistical Analysis

Data are expressed as the standard deviation of the mean for each of the continuous variables, with categorical data being expressed as a percentage of the total patient number. Statistical comparisons were made using the independent *t* test and Fisher’s exact test in order to compare the effects of the continuous variables and categorical variables, respectively. A *p*-value below 0.05 was defined as statistically significant. Logistic regression using univariable and multivariable analysis was applied to determine the factors affecting patient clinical outcomes, as shown by the odds ratio (OR) with a 95% confidence interval (CI). Cox’s regression was used to examine the strength of the association between patient survival and each variable, as shown by the hazard ratio (HR) with a 95% CI. The Kaplan–Meier method was used to analyze the differences between group survival distributions, while the log-rank test was used to assess the differences in survival

## 3. Results

### 3.1. Patient Characteristics and Variables Associated with Non-SVR

Amongst the 97 enrolled patients, 88 and 9 of them achieved SVR and non-SVR to DAA therapy, respectively. The SVR rate was 90.7%. The characteristics of these patients are shown in Table 1. Overall, the mean age was 69.37 years, with a male predominance (67.0%) being noted. The patients with non-SVR undergoing DAA therapy, compared to those with SVR, had a higher ratio of Child–Pugh stage B (33.3% vs. 17.0%, *p* = 0.361), ALBI grade 2/3 (77.8% vs. 56.8%, *p* = 0.316), HCC size (mean 6.09 cm vs. 3.96 cm, *p* = 0.087), beyond up-to-7 criteria (55.6% vs. 36.4%, *p* = 0.295), HCC with PD (44.5% vs. 12.5%, *p* = 0.198), and AFP over 400 ng/mL (33.3% vs. 13.7%, *p* = 0.142), but these differences were all non-significant. The ratio of SVR was high in the HCC patients with CR (19 of 20 cases, 91.6%), PR (19 of 20 cases, 91.6%), and SD (39 of 42 cases, 92.9%), but low in those with PD (11 of 15 cases, 73.3%).

The logistic regression analysis of non-SVR to DAA therapy and the clinical variables are shown in Table 2. The parameters of age, gender, Child–Pugh stage, ALBI grade, BCLC stage, values of AFP, HCV genotype, and pangenotypic DAA regimens had non-significant effects. In contrast, HCC with PD, when compared with others (CR + PR + SD), had a significant impact on non-SVR status (OR 5.59, 95% CI 1.30–24.06, *p* = 0.021).

### 3.2. Patientoverall Survival and the Associated Factors

A total of 32 patients died during the follow-up period of this study. The OS rates of all enrolled subjects at one year and two years after completion of DAA therapy were 85.1% and 68.2%, respectively. The median value of OS was 3.81 years. Further analysis of OS when stratified by the clinical variables is shown in Table 3. The clinical presentations of severe cirrhosis (Child–Pugh stage B vs. A:HR 0.33, 95% CI 0.14–0.80, *p* = 0.014), advanced stage HCC (BCLC stage C vs. B:HR 0.23, 95% CI 0.09–0.58, *p* = 0.020), and viable HCC at the time of DAA therapy initiation (mRECISE PR + SD + PD vs. CR:HR 0.20, 95% CI 0.05–0.68, *p* = 0.010) were all significantly associated with poor survival outcomes. In contrast, achieving SVR, compared with non-SVR, had a positive correlation with OS (SVR vs. non-SVR:HR 8.42, 95% CI 2.93–24.19, *p* = 0.001). The treatments for HCC, including LRT, TKI, and ICI, had no significant effects which affected OS.

The survival outcomes stratified by SVR and non-SVR are shown in Figure 1. The 1-year and 2-year OS rates were significantly higher in the SVR group than those seen in the non-SVR group (1-year: 87.5% vs. 57.1%, *p* = 0.003; 2-year: 71.1% vs. 28.6%, *p* = 0.012). The median OS of the patients with SVR and those with non-SVR to DAA therapy was 4.17 and 1.25 years, respectively. The difference was significant (*p* = 0.008). None of the patients with non-SVR to DAA therapy survived longer than 3 years.

Regarding the patients with SVR, analysis of the association between Child–Pugh stage or BCLC stage and OS is displayed in Figure 2. The patients in Child–Pugh stage A, compared with those who were in Child–Pugh stage B, experienced significantly better OS (*p* = 0.017). The 2-year OS rate in the patients with Child–Pugh stage A and B was 72.1% and 51.1%, respectively (*p* = 0.002). In addition, the patients in BCLC stage B, compared with those who were in BCLC stage C experienced significantly better OS (*p* = 0.006). The 1-year OS rate of the patients with BCLC stage B and C was 91.4% and 60.0%, respectively (*p* = 0.058). None of the patients with BCLC stage C HCC survived more than 2 years.

## 4. Discussion

DAAs are the recommended treatment for patients with HCV infection due to their high rates of eradication, few contraindications, and low rates of adverse events. In patients with advanced cirrhosis, eradication of HCV can lead to improvements in liver function test results, ameliorate hepatic encephalopathy, and reduce the need for liver transplantation [16,17,18]. Moreover, eradication of HCV with DAA reduces the risk of HCC by more than 70% [4].

SVR represents the successful eradication of HCV in patients receiving treatment and has often exceeded 95% in the era of DAA therapy, although its failure rate is higher in subjects with HCC. One meta-analysis showed that pooled SVR rates were significantly lower in patients with HCC than in those without (89.6% vs. 93.3%, *p* = 0.001) [7]. The suboptimal antiviral effect on patients with HCC may be due to ineffective blood delivery to the target site [19] as well as poor cancer immunity impairing viral clearance [20].

One retrospective cohort study investigated 421 HCV patients diagnosed with cirrhosis, of whom 33% had HCC. Failure to achieve SVR was significantly higher in the patients with HCC when compared to those without HCC (21% vs. 12%, *p* = 0.009). The predictor of DAA therapy failure was the presence of active HCC at the time of HCV treatment initiation (OR 8.5, 95% CI 3.90–18.49) [21]. A cohort study performed in Taiwan found that SVR rates were similar in patients with BCLC stage 0/A and B HCC (95% and 97.8%, respectively), but this figure decreased to 77.8% in BCLC stage C HCC patients. For BCLC stage B HCC patients, SVR was higher in the patients without active HCC than in those with active HCC (100% vs. 91.7%) [22].

In our study, which included 97 patients with BCLC stage B/C HCC, there were 9 patients with non-SVR to DAA therapy, and the SVR rate was 90.7%. Further regression analysis found that HCC with PD, or active HCC, at the time of DAA therapy initiation was a significant factor associated with DAA therapy failure (OR 5.59, 95% CI 1.30–24.06, *p* = 0.021). Other clinical parameters, including BCLC stage, had no significant impact on non-SVR status. The SVR rates were similar in patients with BCLC stage B HCC (70/77, 90.9%), as well as those with BCLC stage C HCC (18/20, 90.0%). The higher SVR rate and lack of difference between BCLC stage B and C in our study may be due to a larger proportion of patients having taken high-potency pangenotypic DAAs, such as SOF/VEL (n = 24) and G/P (n = 12).

A prospective study performed in the United States and Canada which analyzed 102 DAA-treated BCLC stage 0/A HCC patients and 102 DAA-untreated BCLC stage 0/A HCC patients using propensity score matching found that OS was significantly higher in the DAA group when compared to the non-DAA group (HR 0.39, 95% CI 0.17–0.91, *p* = 0.03). Furthermore, these data also showed that patients who were likely to have a good outcome from the HCC treatment had a greater likelihood of benefiting from DAA therapy, but patients with advanced HCC, which is known to be associated with poor outcomes, were unlikely to benefit [5]. Another retrospective study which originated in Japan assessed 184 HCV-related HCC patients who had received curative treatment followed by DAA therapy. The findings there showed that eradication with DAAs was an independent factor related to OS (HR 0.32, 95% CI 0.17–0.60, *p* < 0.01) [6]. The use of DAA is related to longer survival times, which may relate to the improvement or preservation of liver function [23]. However, it is not known whether the same results would be achieved in patients with BCLC B/C stage HCC.

A retrospective single-center study in Taiwan enrolled 113 HCV-infected patients with BCLC stage B HCC who had received chemoembolization, with the results determining that DAA-treated patients had longer OS than non-treated patients (40.1 months vs. 22.9 months). Further analysis showed that Eastern Cooperative Oncology Group (ECOG) score, use of DAA, and serum albumin were all significant factors associated with OS [24]. Another population-based retrospective cohort study performed in Taiwan evaluated 1684 HCC patients with HCV infection and concomitant HCC who were treated using sorafenib, with the results revealing that the mean survival times of DAA users were longer than those of non-DAA users (20.7 months vs. 12.5 months) [25].

Amongst our enrolled patients, achieving SVR to DAA therapy was a significantly positive factor affecting OS (SVR vs. non-SVR, HR 8.42, 95% CI 2.93–24.19, *p* = 0.001). The median OS of the patients with SVR and those with non-SVR was 4.17 years and 1.25 years, respectively. Other significant negative factors associated with patient OS in our study included cirrhosis (Child–Pugh stage B vs. A, HR 0.33, 95% CI 0.14–0.80, *p* = 0.014), HCC stage (BCLC stage C vs. B, HR 0.23, 95% CI 0.09–0.58, *p* = 0.020), and viable tumor at the time of DAA therapy initiation (mRECISE PR + SD + PD vs. CR, HR 0.20, 95% CI 0.05–0.68, *p* = 0.010).

It is well known that cirrhosis underlies HCC in most patients and that the functional impairment of the liver has a significant impact on prognosis. One aim of DAA therapy is to avoid liver dysfunction during follow-up, thereby increasing the feasibility of HCC treatment, which ultimately affects survival. However, patients with BCLC stage B/C HCC still experience poor outcomes owing to the development of symptomatic HCC progression prior to cirrhosis complications being resolved, a phenomenon which is known as “irreversibility time-point status” [9]. Unfortunately, there are currently no data available to determine this status.

Our results found that treatments for HCC, including LRT, TKI, and ICI, had no significant effect on OS. This is likely due to the highly variable therapeutic intervals and anti-tumor drug doses our patients were subjected to. Otherwise, the systemic treatments, such as TKI and ICI, are known for being standard treatments for patients with BCLC stage C HCC, which is considered to have a poor survival outcome. In contrast, those with BCLC stage B HCC usually accepted LRT only and had a better OS [26].

Amongst all the 88 patients with SVR in our study, the patients with BCLC stage C HCC when compared to those with BCLC stage B or the patients with Child–Pugh stage A when compared to those with Child–Pugh stage B experienced significantly better OS (*p* = 0.012 and 0.025, respectively). These data indicate that advanced stage HCC and intermediate stage cirrhosis are both associated with poor short-term survival outcomes, even when SVR to DAA therapy has been achieved. For these patients, control of their tumors or cirrhosis-associated complications should be prioritized over receiving HCV eradication therapy [9].

Our study had several limitations. First, this study implemented a retrospective design and was conducted at a single tertiary care center; therefore, selection bias may have existed. A validation study is essential. Second, the subjects diagnosed with Child–Pugh stage C or BCLC stage D HCC who were expected to have extremely poor outcomes were excluded from our study although some of these patients may have benefited owing to their improved baseline liver function. Third, a DAA-untreated control group was lacking. In the DAA era, HCV treatment becomes fast and easy. Therefore, even though the benefits of DAA therapy in advanced HCC have not been completely confirmed, conducting a prospective study to establish a DAA-untreated control group for HCV-related HCC patients may face an ethical challenge in clinical practice. Furthermore, a retrospective study may result in a selection bias in survival analysis, in which HCC patients with a short life-expectancy might not receive DAA therapy. In addition, the systemic therapy for unresectable HCCs has been largely improved in the DAA era; therefore, a historical untreated control group in the interferon era might not be comparable with that in the DAA era. Anyway, the findings of this study may provide a useful basis for further designing a matched control group in the future. Fourth, a small case number in the non-SVR group should have limited a sufficient statistical power in the statistical analysis for identifying the factors related to non-SVR, particularly in the multivariable analysis. However, even though this study is currently one of the large-scale cohort studies investigating DAA therapy for BCLC Stage B/C HCC patients, the high SVR rate of DAA therapy would eventually result in limited non-SVR cases. A multi-center study should be encouraged to resolve this limitation. Lastly, the therapeutic intervals and doses of oncospecific treatments were difficult to analyze in this retrospective study, and those factors may have affected survival outcomes. Further prospective research involving the analysis of more variables is, therefore, warranted.

## 5. Conclusions

In patients diagnosed with BCLC stage B/C HCC, DAA could achieve a high SVR rate except in those patients with PD to previous HCC treatments. SVR was found to be related to improvements in OS, and, therefore, DAA therapy should be encouraged for patients who do not have a short life expectancy.

## Figures and Tables

**Figure 1 viruses-14-02316-f001:**
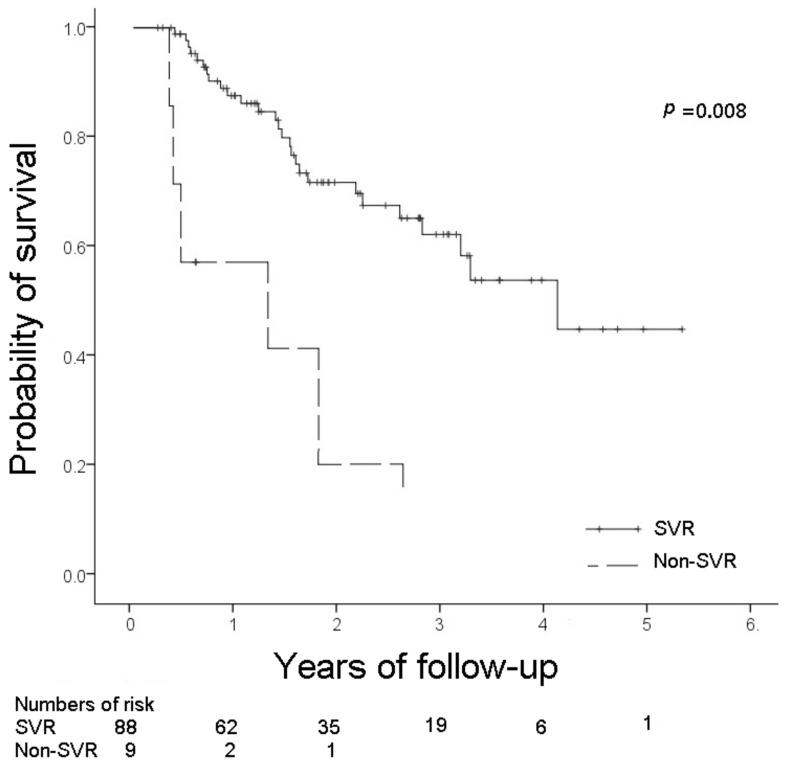
Cumulative probability of survival stratified by SVR and non-SVR (SVR, sustained viral response).

**Figure 2 viruses-14-02316-f002:**
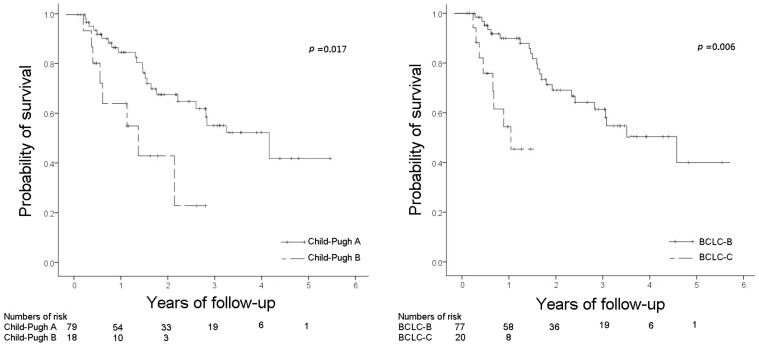
Cumulative probability of survival in the patients with SVR stratified by Child–Pugh stage and BCLC stage (SVR, sustained viral response).

**Table 1 viruses-14-02316-t001:** Baseline characteristics in patients with SVR and non-SVR.

		All (n = 97)			SVR (n = 88)			Non-SVR (n = 9)		
		Mean ± SD	n	%	Mean ± SD	n	%	Mean ± SD	n	%
Age (years)		69.37 ± 10.46			69.13 ± 10.20			71.56 ± 13.29		
Gender	Male		65	(67.0%)		58	(65.9%)		7	(77.8%)
	Female		32	(33.0%)		30	(34.1%)		2	(22.2%)
Child–Pugh stage	A		79	(81.4%)		73	(83.0%)		6	(66.7%)
	B		18	(18.6%)		15	(17.0%)		3	(33.3%)
ALBI grade	1		40	(41.2%)		38	(43.2%)		2	(22.2%)
	2		53	(54.6%)		47	(53.4%)		6	(66.7%)
	3		4	(4.2%)		3	(3.4%)		1	(11.1%)
FIB-4		6.57 ± 6.35			6.65 ± 6.61			5.88 ± 2.91		
Totoal bilirunin (U/L)		0.88 ± 0.54			0.88 ± 0.56			0.88 ± 0.44		
ALT (U/L)		87.49 ± 92.57			85.09 ± 95.77			110.00 ± 49.98		
HBV co-infection			6	(6.2%)		6	(6.8%)		0	
HCV RNA (IU/mL)		5.56 ± 0.99			5.54 ± 0.99			5.82 ± 1.01		
HCV genotype	Ia		5	(5.2%)		4	(4.5%)		1	(11.2%)
	Ib		66	(68.0%)		62	(70.5%)		4	(44.4%)
	II		23	(23.7%)		21	(23.9%)		2	(22.2%)
	III		1	(1.0%)		1	(1.1%)		0	
	VI		2	(2.1%)		0			2	(22.2%)
DAA regimen	DCV/ASV		2	(2.1%)		2	(2.3%)		0	
	EBR/GZR		12	(12.4%)		12	(13.6%)		0	
	SOF/RBV		9	(9.3%)		6	(6.8%)		3	(33.3%)
	SOF/LDV		25	(25.8%)		22	(25.0%)		3	(33.3%)
	SOF/DCV		1	(1.0%)		1	(1.1%)		0	
	SOF/VEL		25	(25.8%)		23	(26.1%)		2	(22.2%)
	PrOD		11	(11.3%)		11	(12.5%)		0	
	G/P		12	(12.4%)		11	(12.5%)		1	(11.1%)
AFP (ng/mL)		1116.1 ± 4415.8			1100.3 ± 4598.3			1272.0 ± 2003.4		
AFP > 400 ng/mL			15	(15.5%)		12	(13.7%)		3	(33.3%)
BCLC stage	B		77	(79.4%)		70	(79.5%)		7	(77.8%)
	C		20	(20.6%)		18	(20.5%)		2	(22.2%)
HCC size (cm)		4.15 ± 3.56			3.95 ± 3.32			6.09 ± 5.31		
Pre-DAA HCC treatment *										
	LRT		80	(82.5%)		73	(83.0%)		7	(77.8%)
	TKI		10	(10.3%)		10	(11.4%)		0	
	ICI		3	(3.1%)		3	(3.4%)		0	
Response to prior HCC treatment										
(mRECIST)	CR		20	(20.6%)		19	(21.6%)		1	(11.1%)
	PR		20	(20.6%)		19	(21.6%)		1	(11.1%)
	SD		42	(43.3%)		39	(44.3%)		3	(33.3%)
	PD		14	(15.5%)		11	(12.5%)		4	(44.5%)
Post-DAA HCC treatment **										
	LRT		89	(91.7%)		82	(93.2%)		7	(77.8%)
	TKI		48	(49.5%)		43	(48.9%)		5	(55.6%)
	ICI		10	(10.3%)		10	(11.4%)		0	

Abbreviations: AFP, alpha-fetoprotein; ALT, alanine aminotransferase; CR, complete response; DAA, direct antiviral agent; HBV, Hepatitis B virus; HCC, hepatocellular carcinoma; ICI, immune checkpoint inhibitor; LRT, local-regional therapy; PD, progressive disease; PR, partial response; SD, standard derivation; SD, stable disease; SVR, sustained viral response; TKI, tyrosin kinase inhibitor. DAA regimens: DCV/ASV, daclatasvir plus asunaprevir; EBR/GZR, elbasvir plus grazoprevir, SOF/RBV, sofosbuvir plus ribavirin; SOF/LDV, sofosbuvir plus ledipasvir; SOF/DCV, sofosbuvir plus daclatasvir; SOF/VEL, sofosbuvir plus velpatasvir; PrOD, ombitasvir and paritaprevir with ritonavir plus dasabuvir; G/P, glecaprevir-pibrentasvir. * HCC treatment before initiating DAA therapy; ** HCC treatments after initiating DAA therapy.

**Table 2 viruses-14-02316-t002:** The strength of association between non-SVR and variables.

	Univariable Analysis		
Variables	OR	(95% CI)	*p*-Value
Age ≤ 65 years old (vs. >65)	0.96	(0.91–1.05)	0.509
Gender male (vs. female)	1.81	(0.35–9.25)	0.576
Child–Pugh stage B (vs. A)	2.43	(0.55–10.83)	0.243
ALBI grade 2/3 (vs. 1)	2.66	(0.52–13.54)	0.238
FIB4 > 3.25 (vs. ≤3.25)	1.55	(0.30–7.95)	0.600
HCV GT1 (vs. other GTs)	0.44	(0.11–1.79)	0.252
HCV GT2 (vs. other GTs)	0.91	(0.18–4.73)	0.912
Non-pangenotypic DAA (vs. pangenotypic DAA)	1.32	(0.31–5.63)	0.707
AFP > 400 (vs. ≤400 ng/mL)	3.17	(0.70–14.38)	0.136
BCLC stage C (vs. B)	1.11	(0.21–5.81)	0.900
Beyond up-to-7 criteria (vs. within)	2.18	(0.55–8.73)	0.268
PR + SD + PD (vs. CR)	2.19	(0.26–18.67)	0.470
SD + PD (vs. CR + PR)	2.66	(0.52–13.54)	0.238
PD (PD vs. CR + PR + SD)	5.59	(1.30–24.06)	0.021

Abbreviations: AFP, alpha-fetoprotein; CR, complete response; DAA, direct antiviral agent; GT, genotype; PD, progressive disease; PR, partial response; SD, stable disease; SOF, sofosbuvir; SVR, sustained viral response.

**Table 3 viruses-14-02316-t003:** The strength of association between overall survival and clinical variables.

	Univariable Analysis			Multivariable Analysis		
Variables	HR	(95% CI)	*p*-Value	HR	(95% CI)	*p*-Value
SVR (vs. no SVR)	5.32	(2.01–14.09)	0.008	8.42	(2.93–24.19)	0.001
Age ≤ 65 (vs. >65 years old)	1.11	(0.52–2.38)	0.781			
Gender male (vs. female)	0.83	(0.38–1.80)	0.640			
Child–Pugh stage B (vs. A)	0.36	(0.15–0.83)	0.017	0.33	(0.14–0.80)	0.014
ALBI grade 2/3 (vs. 1)	0.61	(0.29–1.26)	0.185			
FIB4 > 3.25 (vs. ≤3.25)	0.79	(0.35–1.78)	0.579			
HCV GT1 (vs. other GTs)	1.09	(0.48–2.45)	0.832			
HCV GT2 (vs. other GTs)	1.16	(0.49–2.72)	0.722			
PangenotypicDAA (vs. other DAA)	1.06	(0.25– 4.52)	0.367			
AFP > 400 (vs. ≤400 ng/mL)	0.53	(0.21–1.31)	0.175			
BCLC stage C (vs. B)	0.29	(0.12–0.70)	0.006	0.23	(0.09–0.58)	0.020
Beyond up-to-7 criteria (vs. within)	0.51	(0.25–1.07)	0.007			
Pre-DAA LRT (vs. noLRT)	2.71	(0.91–6.62)	0.282			
Pre-DAA TKI(vs. noTKI)	0.81	(0.19–3.47)	0.784			
Pre-DAA ICI (vs. noICI)	0.39	(0.05–3.01)	0.367			
PR + SD + PD (vs. CR)	0.21	(0.06–0.71)	0.012	0.20	(0.05–0.68)	0.010
SD/PD (vs. CR + PR)	0.51	(0.24–1.08)	0.081			
PD (vs. CR + PR + SD)	0.68	(0.26–1.78)	0.441			
Post-DAA LRT(vs. noLRT)	2.40	(0.72–8.02)	0.152			
Post-DAA TKI (vs. noTKI)	0.52	(0.25–1.05)	0.070			
Post-DAA ICI (vs. noICI)	0.87	(0.26–2.89)	0.825			

Abbreviations: AFP, alpha-fetoprotein; CR, complete response; DAA, direct antiviral agent; GT, genotype; ICI, immune checkpoint inhibitor; LRT, local-regional therapy; PD, progressive disease; PR, partial response; SD, stable disease; SVR, sustained viral response, TKI, tyrosin kinase inhibitor.

## Data Availability

The data presented in this study are available on reasonable request to the corresponding author.

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
