# Peer review of "Direct-Acting Antiviral Therapy for Hepatitis C Virus in Patients with BCLC Stage B/C Hepatocellular Carcinoma"

_viruses, 2022, doi:10.3390/v14112316_

Round 1
Reviewer 1 Report
Lee et al present data on a clinical relevant topic. However, the cohort is overall small and the value limited by the small number of events. There are major flaws in the statistical analysis. Thus, conclusion are not supported by the results.
Major Comments:
1. I can not find a statement regarding an ethical approval of the study
2. Die the patients provide written informed consent for the use of their data?
3. Statistical analysis are inappropriate. Given the low number of evets a multivariable analysis can not be performed. Usually 6-8 events are required per variable included in a multivariable model.
4. Unvariable analysis is not controlled for multiple testing.
5. Co-linearity between CHILD an BCLC needs to be considered.
6. How was a patient handled that died during DAA therapy? Censored? Counted as non-SVR?
Minor comments:
1. Results: Median age was 69.37? Mean age?
Author Response
Comment 1: I can not find a statement regarding an ethical approval of the study.
Response: Thank you for your kind suggestion. According to the submission manuscript form of Viruses, the ethical approval of this study has been listed in Institutional Review Board Statement(Page 11, Lines 328-330)as follows: The study was conducted in accordance with the Declaration of Helsinki, and approved by the Institutional Review Board of Taichung Veterans General Hospital(protocol code CE21059B-1 and approval on 2020/12/15).
Comment 2: Did the patients provide written informed consent for the use of their data?
Response:Thank you for your kind suggestion. According to the submission manuscript form of Viruses, the ethical approval of this study has been listed in Informed Consent Statement(Page 11, Lines 331-332)as follows: The need for written informed consent was waived by the Ethical Review Board of Taichung Veterans General Hospital(protocol code CE21059B-1).
Comment 3: Statistical analysis are inappropriate. Given the low number of evets a multivariable analysis can not be performed. Usually 6-8 events are required per variable included in a multivariable model.
Response: Thank you for this constructive comment. We agree with you that the event numberis small in the logistic regression analysis for variables associated with non-SVR (Table 2; n= 9), therefore we have revised the statistical analysis according to your constructive comment.In the univariable regression analysis, only one variable (PD vs. CR+PR+SD; Table 2) had a significant impact on non-SVR status (OR 5.59, 95% CI 1.30-24.06, P=0.021), therefore we deleted the multivariable analysis in Table 2. In addition, we have further addressed the limitations of the small case number of non-SVR in the study limitation paragraph (Page 10, Lines 301-307). However, in the Cox’s regression analysis for patient overall survival, a total of 32 patients died during the follow-up period of this study. As shown in Table 3, four variables (Child-Pugh stage B, BCLC stage C, Beyond up-to-7 criteria, and SVR) were significantly associated with overall survival in the univariable analysis, therefore the event number (death; n= 32) should have been sufficient for a further multivariable regression analysis. The related statements have been addressed in the manuscript (Page 6, Line151).
Comment 4: Unvariable analysis is not controlled for multiple testing.
Response: Thank you for this importantcomment. A small case number in the non-SVR group (n= 9) should have limited a sufficient statistical power inthe statistical analysis for identifying the factors related to non-SVR, particularly in the multivariable analysis.However, in the univariable regression analysis, only one variable (PD vs. CR+PR+SD; Table 2) had a significant impact on non-SVR status (OR 5.59, 95% CI 1.30-24.06, P=0.021), therefore we did not perform the multivariable analysis in Table 2. Even though this study is currently one of the large-scale cohort studies investigating DAA therapy for BCLC Stage B/C HCC patients, the high SVR rate of DAA therapy would eventually result in limited non-SVR cases. A multi-center study should be encouraged to resolve this limitation. We have further addressed the limitation of the small case number of non-SVR in the study limitation paragraph (Page 10, Lines 301-307).
Comment 5: Co-linearity between CHILD an BCLC needs to be considered.
Response: Thank you for this carefulconsideration. As shown in Table 3, the univariable regression analysis has been conductedfor exploring the possible prognostic factors in overall patient survival, and 4 variables (Child-Pugh stage B, BCLC stage C, Beyond up-to-7 criteria, and SVR) are significantly associated with overall survival. Because a total of 32 patients died during the follow-up period of this study, the event number should have been sufficient for the further multivariable regression analysis to avoid the possible co-linearity among the variables. In the multivariable regression analysis, Child-Pugh stage B, BCLC stage C, Beyond up-to-7 criteria, and SVR remained independently associated with overall survival (Table 3; Page 6, Lines 151-162).
Comment 6: How was a patient handled that died during DAA therapy? Censored? Counted as non-SVR?
Response: Thank you for this helpful query. For avoiding an immortal time bias in outcome analysis, only patients who completedDAA therapy with available SVR datawere included in this study, therefore no patient died during DAA therapy.SVR was defined as undetectable HCV RNA at least 12 weeks (SVR12) after completion of DAA therapy. The related statements have been clarified in the manuscript (Materials and Methods: Page 2, Lines 68-72; Page 3, Lines 105-106).
Comment 7: Median age was 69.37? Mean age?
Response: Thank you for this helpful query. The mean age was 69.37 years. The related statements have been corrected in the manuscript (Table 1; Page 3, Line 127).
Reviewer 2 Report
This single-center study aimed at evaluating the benefit of HCV eradication in patients with active HCC in intermediate-advanced stage. A total of 97 patients with BCLC B/C HCC were retrospectively evaluated for survival outcomes in order to assess the potential benefit of DAA-based treatment.
The aim of the study is a field of great interest, as DAA-induced gain in liver function could provide a survival benefit by improving liver function and making patients suitable for more aggressive (and prolonged) HCC treatment strategies. However, the present study has remarkable methodologic flaws that prevent the implementation of study aims.
1) Methods: the study is retrospective and the enrolment period is extremely spanned (2009-2021), so that progressive change in HCC treatments (i.e. new therapeutic regimens available) could significantly impact study results.
Moreover, in order to really assess the benefit of SVR on survival in patients with active HCC, the correct comparator should be an untreated control group and not non-SVR patients. The benefits of DAA-treatment in advanced HCC are currently not established, so that it could be ethically correct to include untreated patients (in a prospective manner) or retrospectively compare outcomes with patients not receiving DAA-therapy through a correct patient matching (i.e. propensity score analysis).
2) Results: Patients characteristics in Table 1 should be better provided for all patients together (and then by dividing SVR vs non-SVR patients)
3) Results: Comparisons between 9 non-SVR patients vs. 88 SVR patients are extremely underpowered from a statistical point of view in order to detect an association with survival outcome
4) Results: Survival figures for BCLC B and C patients are respectively 4 and 1.8 months. These figures seem to be extremely low. More details concerning HCC treatments (timing and type) should be provided.
Author Response
Comment 1: Methods: the study is retrospective and the enrolment period is extremely spanned (2009-2021), so that progressive change in HCC treatments (i.e. new therapeutic regimens available) could significantly impact study results.
Response: Thank you for your careful review. We mistakenly included the period of interferon era in the treatment of chronic HCV infection, and the study enrollment period should be corrected as follows: Data on patients with HCC, BCLC stage B or Cat Taichung Veterans General Hospital during the period from January 2017 to March 2021, were screened retrospectively. Therefore, we considered that the change in HCC treatments should be acceptable in this limited study period (Abstract: Page 1, Line20; Materials and Methods: Page 2, Line 67)
Comment 2:Moreover, in order to really assess the benefit of SVR on survival in patients with active HCC, the correct comparator should be an untreated control group and not non-SVR patients. The benefits of DAA-treatment in advanced HCC are currently not established, so that it could be ethically correct to include untreated patients (in a prospective manner) or retrospectively compare outcomes with patients not receiving DAA-therapy through a correct patient matching (i.e. propensity score analysis).
Response: Thank you for this important comment.We totally agree with you that a DAA-untreated control group should be the perfect comparator to assess the benefit of DAA therapy on patient survival; however, some concerns may also be raised on establishing an DAA-untreated control group: First, in the era of DAA therapy, HCV treatment becomes fast and easy. Therefore, even though the benefits of DAA therapy in advanced HCC have not been completely confirmed, conducting a prospective study to establish a DAA-untreated control group for HCV-related HCC patientsmayface anethical challenge in clinical practice. Second, a retrospective study may result in a selection bias in survival analysis, in which HCC patients with a short life-expectancy might not receiveDAA therapy. Third, the systemic therapy for unresectable HCCshas been largely improved in the era of DAA therapy, therefore a historical untreated control group in the era of interferon therapy might not be comparable with that in the era of DAA therapy. In this study, some independent prognostic factors were identified via the multivariable regression analysis, and the findings of this study may provide a useful basis for further designing a matched control group in the future. We have discussed the above-mentioned concerns in the Limitation paragraph of the manuscript (Page 10, Lines 291-301).
Comment 3:Results: Patients characteristics in Table 1 should be better provided for all patients together (and then by dividing SVR vs non-SVR patients).
Response: Thank you for this constructive suggestion. The data of patient characteristics for all patients together have been provided in Table 1.
Comment 4:Results: Comparisons between 9 non-SVR patients vs. 88 SVR patients are extremely underpowered from a statistical point of view in order to detect an association with survival outcome.
Response: Thank you for this helpful comment. We agree with you that a small case number in the non-SVR group should have a limited statistical power in the statistical analysis for survival outcome, particularly in the multivariable analysis. However, in the Cox’s regression analysis for patient overall survival, a total of 32 patients died during the follow-up period of this study. As shown in Table 3, four variables (Child-Pugh stage B, BCLC stage C, Beyond up-to-7 criteria, and SVR) were significantly associated with overall survival in the univariable analysis, therefore the event number (death; n= 32) should have been sufficient for a further multivariable regression analysis. Even though this study is currently one of the large-scale cohort studies investigating DAA therapy for BCLC Stage B/C HCC patients, the high SVR rate of DAA therapy would eventually result in limited non-SVR cases. A multi-center study should be encouraged to resolve this limitation. We have further addressed the limitation of the small case number of non-SVR in the study limitation paragraph (Page 10, Lines 301-307).
Comment 5:Results: Survival figures for BCLC B and C patients are respectively 4 and 1.8 months. These figures seem to be extremely low. More details concerning HCC treatments (timing and type) should be provided.
Response: Thank you for this helpful comment. The survival data for BCLC B and C patients were mistakenly described, and “years” should be the correct unit instead of “months”. We have revised the statements in the manuscript (through Page 7, Line 178, toPage 8, Line 192). Furthermore, we have also provided the details (timing and type) of HCC treatmentsin the manuscript, including locoregional treatment (LRT), tyrosine kinase inhibitor (TKI), and immune checkpoint inhibitor (ICI), and the impacts of various HCC treatments were included in the regression analysis for patient overall survival (Table 1, Table 3;through Page 2, Line 91, to Page 3, Line 92).
Reviewer 3 Report
S-H Lee and colleagues describe in their manuscript entitled "Direct-acting Antiviral Therapy for Hepatitis C Virus in Patients with BCLC Stage B/C Hepatocellular Carcinoma" the effect of anti-HCV therapy in patients with advanced HCC on the sustained virological response and overall survival. The manuscript is well written and highly relevant.
General comment: The target audience of the journal Viruses are virologists and might not be familiar with some of clinical oncology terms and concept used in this manuscript. The readers might benefit from an explanation of these concepts and terms in the introduction.
Minor comments:
line 18: please define abbreviation "BCLC"
line 52: as above, please define abbreviation "BCLC" and explain significance of this classification system
lines 67-70: please explain significance of clinical oncology parameters such as "child-Pugh stage, ALBI grade, FIB-4 score etc
line 74: as mentioned above, readers might be unfamiliar with "up-to=7 criteria"
Table 1: 8 out of 9 patients with-out SVR were on a sofosbuvir-containing regimen, while 36 out 36 patients on the other DAA regimens achieved SCR. Please discuss.
Author Response
Comment 1: General comment: The target audience of the journal Viruses are virologists and might not be familiar with some of clinical oncology terms and concept used in this manuscript. The readers might benefit from an explanation of these concepts and terms in the introduction.
Response: Thank you for this constructive suggestion. We have thoroughly reviewed and explained the clinical oncology terms and concept used in the manuscript, including BCLC stage, Child-Pugh stage, ALBI grade, FIB-4 score, and up-to-7 criteria (Page 2, Lines 49-53; Page 2, Lines 77-83; Page 2, Lines 89-91; Page 3, Lines 94-95).
Comment 2: line 18: please define abbreviation "BCLC".
Response: The abbreviation of BCLC has been defined in full form as Barcelona Clinic Liver Cancer (Page 1, Line 18).
Comment 3: line 52: as above, please define abbreviation "BCLC" and explain significance of this classification system.
Response: The abbreviation of BCLC has been defined in full form as Barcelona Clinic Liver Cancer. The definition of HCC in BCLC stage B or C is equal to intermediate or advanced HCC. In addition, BCLC stage B/C HCC, i.e. multiple large tumors, tumors with vascular invasion, and/or extrahepatic metastasis, is basically incurable, and its patient survival time is much shorter than that of BCLC stage A (early HCC).We have explained the clinical significance of this BCLC classification system in the manuscript (Page 2, Lines 49-53; Page 3, Lines 94-95).
Comment 4: lines 67-70: please explain significance of clinical oncology parameters such as "child-Pugh stage, ALBI grade, FIB-4 score etc.
Response: Child-Pugh stage A/B/C was evaluated by the composite parameters (albumin, bilirubin, prothrombin time, ascites, and hepatic encephalopathy) to assess the severity of chronic liver disease. ALBI grade 1/2/3 was calculated only by albumin and bilirubin, which was another useful parameter to access liver dysfunction. Without liver biopsy, FIB-4 scores were calculated by ALT/ aspartate aminotransferase/ platelet/ age, which was used to evaluate the degree of liver fibrosis. We have explained the clinical significance of these parameters in the manuscript (Page 2, Lines 77-83).
Comment 5: line 74: as mentioned above, readers might be unfamiliar with "up-to=7 criteria".
Response: Thank you for this constructive suggestion. The up-to-7 criteria means HCC with seven as the sum of the diameter of the largest tumor (in cm) and the number of tumors, and HCC beyond the up-to-7 criteria indicates a large tumor burden (Page 2, Lines 89-91).
Comment 6: Table 1: 8 out of 9 patients without SVR were on a sofosbuvir-containing regimen, while 36 out 36 patients on the other DAA regimens achieved SVR. Please discuss.
Response: Thank you for this constructive comment. After carefully reviewing SOF-based DAA regimens used in this study, we found that the SOF-based DAA regimens werequite heterogeneous and might not be completely indicated as a strong-potency antiviral therapy. For example, the SVR rate in sofosbuvir plus ribavirin (SOF/RBV) therapy was lower than that in other pangenotypic DAA therapies, such as a non-SOF-based regimen, glecaprevir-pibrentasvir (G/P). Therefore, SOF-based DAAmay not be a good way to classify the potency of DAA therapy. We thus re-classified the DAA regimens into pangenotypic or non-pangenotypic regimens, and sofosbuvir plus daclatasvir (SOF/DCV), sofosbuvir plus velpatasvir (SOF/VEL), and glecaprevir-pibrentasvir (G/P) weredefined as pangenotypic DAAs in this study. Among patients without SVR, 3 out of 9 patients received pangenotypic DAAs. However, in the regression analysis, non-pangenotypic DAA was not significantly associated with non-SVR (OR 1.32; 95% CI: 0.31-5.63), and only PD to prior HCC treatment was associated with non-SVR (OR 5.59; 95% CI: 1.30-24.06). This result might be explained by the appropriate selection for a non-pangenotypic regimen according to its indicated genotypes. A large-scale study should be important to validate our findings. We have revised the related statements, statistical data, and discussion in the manuscript (Table 2; Table 3; Page 3, Lines 103-104; Page 9, Lines 229-231).
Round 2
Reviewer 2 Report
Authors have addressed the majority of issues raised